# Lipocalin-2, Matrix Metalloproteinase-9, and MMP-9/NGAL Complex in Upper Aerodigestive Tract Carcinomas: A Pilot Study

**DOI:** 10.3390/cells14070506

**Published:** 2025-03-29

**Authors:** Luca Cavalcanti, Silvia Francati, Giampiero Ferraguti, Francesca Fanfarillo, Daniele Peluso, Christian Barbato, Antonio Greco, Antonio Minni, Carla Petrella

**Affiliations:** 1Department of Sensory Organs, Sapienza University of Rome, 00161 Roma, Italy; luca.cavalcanti@uniroma1.it (L.C.); antonio.greco@uniroma1.it (A.G.); antonio.minni@uniroma1.it (A.M.); 2Department of Experimental Medicine, Sapienza University of Rome, 00161 Roma, Italy; silvia.francati@uniroma1.it (S.F.); giampiero.ferraguti@uniroma1.it (G.F.); francesca.fanfarillo@uniroma1.it (F.F.); 3PhD School of Applied Medical-Surgical Sciences, University of Rome “Tor Vergata”, Via Montpellier 1, 00133 Roma, Italy; daniele.peluso@gmail.com; 4Department of Biology, University of Rome “Tor Vergata”, 00133 Roma, Italy; 5Institute of Biochemistry and Cell Biology (IBBC-CNR), 00161 Roma, Italy; christian.barbato@cnr.it; 6Division of Otolaryngology-Head and Neck Surgery, San Camillo de Lellis Hospital, ASL Rieti-Sapienza University, Viale Kennedy, 02100 Rieti, Italy

**Keywords:** lipocalin-2, matrix metalloproteinase-9, MMP-9/NGAL complex, upper aerodigestive tract carcinomas

## Abstract

Upper aerodigestive tract (UADT) carcinomas have a high and rapidly increasing incidence, particularly in industrialized countries. The identification of diagnostic and prognostic biomarkers remains a key objective in oncological research. However, conflicting data have been reported regarding Lipocalin-2 (LCN-2 or NGAL), Matrix Metalloproteinase-9 (MMP-9), and the MMP-9/NGAL complex in UADT carcinomas. For this reason, the primary aim of this study was to investigate the involvement and modulation of the LCN-2 system in UADT cancer by selecting patients at first diagnosis and excluding any pharmacological or interventional treatments that could act as confounding factors. In this clinical retrospective pilot study, we investigated *LCN-2* and *MMP-9* tissue gene expression, as well as circulating levels of LCN-2, MMP-9, and the MMP-9/NGAL complex. Our findings revealed a downregulation of *LCN-2* and an upregulation of *MMP-9* gene expression in tumor tissues compared to healthy counterparts. A similar trend was observed in circulating levels, with decreased LCN-2 and increased MMP-9 in cancer patients compared to healthy controls. Additionally, serum levels of the MMP-9/NGAL complex were significantly elevated in UADT cancer patients relative to controls. Our study suggests a potentially distinct role for the free form of LCN-2 and its conjugated form (MMP-9/NGAL complex) in UADT tumors. These findings not only provide new insights into the molecular mechanisms underlying tumor progression but also highlight the potential clinical relevance of these biomarkers. The differential expression patterns observed suggest that the LCN-2 and MMP-9/NGAL complex could serve as valuable tools for improving early diagnosis, monitoring disease progression, and potentially guiding therapeutic strategies. Further research is needed to validate their utility in clinical settings and to explore their prognostic and predictive value in personalized treatment approaches.

## 1. Introduction

Cancers of the upper aerodigestive tract (UADT) involve the oral cavity (lips, tongue, floor of the mouth, gingival mucosa, inner surface of the cheeks and palate), the nasopharynx, the oropharynx, the hypopharynx, and the larynx [1]. UADT cancers rank fourth in terms of cancer incidence and second in terms of cancer mortality worldwide [2,3]. Advanced forms are more difficult to cure even after therapeutic care, although localized forms can usually be managed in the long term with surgery or treatments (chemotherapy, radiotherapy, and/or immunotherapy). The primary risk factor for UADT cancer is prolonged exposure to alcohol and tobacco [4,5,6], often associated with an unfavorable prognosis. The identification of diagnostic and prognostic biomarkers remains a crucial objective for oncological research to early identify patients who respond or not to targeted therapies or who may or may not develop metastases or recurrences [7,8,9,10,11,12].

Lipocalin-2 (LCN-2), also referred to as oncogene 24p3 or neutrophil gelatinase-associated lipocalin (NGAL), is a small glycoprotein weighing 25 kDa, and it belongs to the lipocalin family. Initially, LCN-2 was isolated from activated human neutrophils and found to be associated with gelatinase, an enzyme crucial for neutrophil diapedesis [13,14,15]. It has subsequently been shown that NGAL was stored in specific neutrophil granules, not only as a heterodimer with gelatinase but also as a monomer or homodimer [16]. LCN-2 plays a role in various biological functions, particularly in pathological conditions [13,17,18]. Specifically, LCN-2 regulates iron trafficking during bacterial infections, mediating antibiotic activity by sequestering siderophore-iron delivery, which is vital for the survival of microorganisms. Additionally, LCN-2 can influence iron homeostasis in malignant cells, impacting cell proliferation, angiogenesis, migration, and invasion across different cancer types [17,19]. Elevated tissue levels of LCN-2 have been found in several types of cancers [18], linked to increased cell proliferation and metastasis [18]. However, recent research indicates that LCN-2 is significantly down-regulated in primary and metastatic malignant tumors of the oral cavity compared to healthy tissues [20]. Moreover, the downregulation of LCN-2 appears to correlate with the degree of differentiation and the stage of the disease, suggesting a prognostic role for oral cancer [20].

Matrix metalloprotease 9 (MMP-9), also known as gelatinase B, is a zinc metalloproteinase that plays a role in breaking down extracellular matrix components and the basement membrane; high levels of MMP-9 are associated with a greater capacity for neoplastic invasion [21,22].

MMP-9 and LCN-2 can form a protein complex (MMP-9/NGAL complex) that protects MMP-9 from degradation, thereby extending its enzymatic activity [23,24]. Therefore, this interaction plays a fundamental role in the process of progression and metastasis in many neoplasms [25,26].

In the present clinical retrospective observational study, we analyzed *LCN-2* and *MMP-9* tissue gene expression and LCN-2, MMP-9, and MMP-9/NGAL complex serum levels in patients with a first diagnosis of UADT cancer and avoided any kind of treatment (pharmacological and/or interventional) and correlated the results with the observed hematological profile.

## 2. Materials and Methods

### 2.1. Participants’ Selection and Sampling

In this pilot study, we recruited 30 adult individuals (24 men and 6 women) suffering from UADT carcinomas over a period of 2 years (2020–2021) who attended the Otolaryngology Clinic of the “Policlinico Umberto I”, Sapienza University Hospital. As inclusion criteria, we recruited adult male and female patients entering the hospital for the first diagnosis of UADT carcinomas. Based on the power analysis calculator for the sample size in a pilot study, we obtained that with at least 15 samples per group (30 total sample size), we could guarantee an actual power of 0.91. Greater test power has been obtained with the addition of just a few more samples (0.95 for a 35 total sample size), which indeed occurred during the recruitment of both patients and healthy individuals (for additional details, see Appendix A).

Regarding the exclusion criteria, we did not include patients with cancer relapse, those undergoing drug treatments (such as chemotherapy, anti-inflammatory drugs, or immunosuppressants), or those receiving radiotherapy and/or immunotherapy. Additionally, patients with severe infectious diseases (HIV, HBV, or HCV) or other ongoing inflammatory, cardiovascular, endocrine, or autoimmune disorders were excluded. Patients were enrolled at the time of diagnosis, and all samples were collected during pre-hospitalization, prior to the initiation of any treatment. Moreover, 20 adult healthy controls, age- and sex-matched, were selected among the blood donors from the Policlinico Umberto I Transfusion Center. Moreover, based on questionnaires administered during the recruitment phase, both controls and patients reported being smokers and occasional alcohol consumers.

Peripheral blood samples (5 mL) were collected in BD Vacutainer™ Serum Separation Tubes and centrifuged at 3000 rpm for 15 min to separate the serum from blood cells. The serum was then stored at −80 °C until the day of analysis.

Tumor tissues (n = 9) and non-tumor tissues (n = 19), collected from the resection margins during surgery, were obtained from patients eligible for tumor removal. These tissues were used for RNA extraction to analyze the gene expression of interest (see above).

The study was approved by the Ethics Committee of the “Policlinico Umberto I” Hospital (Rif. 6462); informed consent was signed by each patient, and all the study procedures were in accordance with the Helsinki Declaration of 1975, as revised in 1983, for human experimentation.

### 2.2. Hematologic Parameters

Hematologic parameters were evaluated in the laboratory of the Experimental Medicine Department of the Policlinico Umberto I hospital. Serum glycemia (70.3–100.9 mg/dL), albumin (35–55 g/L), PCR (100–6000 μg/L), transferrin (2.15–3.65 g/L), protein (60–82 g/L), urea (10.2–49.8 mg/dL), phosphate (2.80–4.60 mg/dL), magnesium (0.66–1.03 mmol/L), calcium (2.1–2.5 mmol/L), chloride (98–109 mmol/L), potassium (3.4–5.5 mmol/L), sodium (136–145 mmol/L), iron (64.8–174.9 µg/dL), ferritin (30–400 μg/L), lactate dehydrogenase LDH (males 135–225 UI/L; females 135–214 UI/L), aspartate aminotransferase AST (males 9–45 UI/L; females 9–35 UI/L), alanine aminotransferase ALT (males 10–40 UI/L; females 7–35 UI/L), gamma glutamyl transpeptidase gGT (males 8–61 UI/L; females 5–36 UI/L), direct bilirubin (<0.20 mg/dL), total bilirubin (0.3–1.2 mg/dL), creatinine (males 0.7–1.2 mg/dL; females 0.5–0.9 mg/dL) and creatinine kinase CPK (males 20–200 UI/L; females 20–180 UI/L) were analyzed using Cobas C 501 platform (Roche Diagnostics, Mannheim, Germany).

### 2.3. RNA Extraction

Approximately 10 mg of tumor and non-tumor tissues were obtained for each patient, the latter taken from the margins of the resection. Tissue samples were immediately immersed in 1 mL of Trizol reagent and stored at −80 °C until the moment they were processed. Tissues were homogenized before extraction; RNA was obtained using Trizol Reagent (Invitrogen, Waltham, MA, USA) protocol, following manufacturer instructions. RNA from each available tissue was obtained in a volume of 20 μL. Extracted RNA was then quantified using the Qubit fluorimetric assay (Invitrogen, Waltham, MA, USA).

### 2.4. Reverse Transcription

A DNase treatment was performed to remove any contaminating genomic DNA: 4 μg of total RNA were incubated with 2.2 units (0.7 μL) of DNase I (New England Biolabs, Ipswich, MA, USA) and 2 μL of 10x DNase1 reaction buffer at 37 °C for 10′ in a final volume of 20 μL. Next, 2.2 μL of 1x EDTA (5 mM, pH = 8) (Sigma-Aldrich, St. Louis, MO, USA) were added to the samples; a 75 °C incubation for 10′ was then performed for enzyme inactivation. From the digested sample, 5.5 μL (corresponding to 1 mg of RNA) was added to 1 μL of iScript Reverse Transcriptase, 4 μL of 5x iScript Reaction mix, and 9.5 μL of distilled water (all reagents from iScript cDNA Synthesis Kit—Bio-Rad, Hercules, CA, USA) in a final volume of 20 μL. The reaction was placed in a PTC100 thermal cycler (Bio-Rad, Hercules, CA, USA) and followed the protocol: 25 °C for 5′; 42 °C for 30′; 85 °C for 5′. As quality control, an amplification of the GAPDH housekeeping gene was performed: 2 μL of cDNA, 5.7 μL of distilled water, 2.1 μL of dNTPs 1.25 mM (Thermo Fisher Scientific, Waltham, MA, USA), 3 μL of Buffer Mix Reaction 5x (Promega, Madison, WI, USA), 0.9 μL MgCl_2_ 25 mM (Promega, Madison, WI, USA), 0.1 μL di Gotaq DNA polymerase 5 U/mL (Promega, Madison, WI, USA), 0.6 μL of the GAPDH forward primer (5′-CCCTTCATTGACCTCAACTACATG-3′) and 0.6 μL of the reverse primer (5′-TGGGATTTCCATTGATGACAAGC-3′) in a total reaction volume of 15 μL were assembled. Thermal cycling was set as follows: 95 °C for 2′, (94 °C for 45″; 60 °C for 1.5′; 70 °C for 2.5′) for 28 cycles; 72 °C for 7′; 10 °C ∞. Electrophoresis (3.3 μL of each sample) was performed in a 2% agarose gel (Bio-Rad, Hercules, CA, USA); PCR amplicons were distinguished by their size using a 50 bp molecular weight standard (Thermo Fisher, Waltham, MA, USA).

### 2.5. Digital Droplet PCR

Digital droplet PCR (ddPCR) assays were performed by assembling 1 μL of cDNA mix with 10 μL of 2X ddPCR Probe Supermix (no dUTP) (Bio-Rad, Hercules, CA, USA, 1863024), 1 μL of TaqMan LCN-2 gene expression probe (code 4331182, ID Hs01008571_m1; ThermoFisher Scientific) or MMP-9 (code 4453320, ID Hs00957562_m1; ThermoFisher Scientific), and 8 μL of H_2_O in a final reaction volume of 20 μL [27]. For details on the procedure, see the Appendix A.

### 2.6. Serum LCN-2, MMP-9, and MMP-9/NGAL Complex Analysis

Human serum LCN-2 (Cat. No. DY8556), MMP-9 (Cat. No. DY911), and MMP-9/NGAL complex (Cat. No. DY1757) were measured using sandwich enzyme-linked immunosorbent assay (ELISA) kits (DuoSet ELISA, R&D Systems, Minneapolis, MN, USA), according to the protocols provided by the manufacturer. For details on the procedure, see the Appendix A.

### 2.7. Statistical Analysis

The statistical analysis was performed using Prism 9 for MacOS, Version 9.3.1. Data were presented as mean ± standard deviation (SD) or standard error mean (SEM), where appropriate. Statistical significance was determined using one-way ANOVA, two-way ANOVA, and Student’s *t*-test where appropriate. Receiver operating characteristics (ROC) analyses were performed with the standard parameters in Prism 9, using the Wilson/Brown method for confidence interval calculation. For all tests, the threshold of statistical significance was 0.05. The Spearman correlation test was performed to investigate the association between serum LCN-2, MMP-9, and MMP-9/NGAL complex levels with the hematological parameters measured.

## 3. Results

### 3.1. General Description of the Enrolled Individuals

The cohort of cancer patients enrolled in the study (n. 30 with a middle age of 62.5 ± 1.7 years old) and of healthy controls (n. 20 with a middle age of 58.7 ± 1.0 years old) showed a higher prevalence of men compared to women in both groups (80% vs. 20%). This distribution aligns with global epidemiological data, which indicate that UADT cancers are more common in men than in women [28]. The general characteristics of the enrolled patients have been summarized in Table 1. The clinical (c) or pathological (p) tumor-node-metastasis TNM staging system followed the AJCC/UICC—TNM staging system—8th Edition [29,30].

### 3.2. Hematologic Parameters

Table 2 illustrates the hematologic parameters detected in recruited patients, which resulted in significantly different results compared to healthy controls. Overall, oncologic patients showed a very similar hematologic profile, and some alterations were in line with the pathological condition, such as neoplastic cachexia, inadequate caloric-protein consumption, and chronic inflammation state. In particular, they displayed glucose, urea, and phosphate increases as well as augmented ferritin, direct bilirubin, gGT, and PCR. Conversely, a decrease in albumin, total protein, iron, transferrin, creatinine, LDH, and CPK was found compared to healthy controls.

### 3.3. Tissue LCN-2 and MMP-9 Gene Expression

The available tissue analysis revealed that in tumoral samples, *LCN-2* had a lower (t = 2.002, df = 26, *p* = 0.056) copy number with respect to normal tissues. On the contrary, *MMP-9* expression in tumoral portions was significantly higher with respect to peri-tumoral tissues (t = 2.996, df = 26, *p* = 0.0059) (Figure 1A,B). When analyzing peri-tumoral and tumoral tissues from the same UADT cancer patients (n = 9), we observed a similar, though not significant, trend in the expression of both *LCN-2* and *MMP-9* (Figure 1A’,B’).

### 3.4. Serum LCN-2, MMP-9, and MMP-9/NGAL Complex

As shown in Figure 2, LCN-2 serum levels were decreased in UADT cancer patients in comparison to healthy controls (t = 4.796, df = 38, *p* < 0.0001). Conversely, MMP-9 and MMP-9/NGAL complex circulating levels were up-regulated in the same groups (MMP-9: t = 2.897, df = 42, *p* = 0.006; MMP-9/NGAL complex: t = 3.447, df = 42, *p* = 0.001).

In the present pilot study, no significant differences were found in any of the three biomarkers analyzed when UADT patients were stratified by distinct disease stages or lymph node involvement (see Appendix A in Appendix A).

ROC curve analysis (Figure 3) revealed the ability of LCN-2 to discriminate between healthy controls and UADT cancer patients, surpassing the other biological markers tested (MMP-9 and MMP-9/NGAL complex).

### 3.5. Spearman’s Correlation

Table 3 summarized only the statistically significant correlations between serum LCN-2, MMP-9, and MMP-9/NGAL complex and hematologic parameters in healthy controls and UADT cancer patients. Interestingly, in physiological conditions, LCN-2 is positively correlated with transferrin and negatively with ferritin, both proteins involved in iron trafficking. The conjugated form of LCN-2 (MMP-9/NGAL complex) is directly correlated to MMP-9 serum levels. In cancer patients, only the inverse relation between LCN-2 and ferritin is conserved, even if weaker than in healthy conditions, whilst a strong direct association with MMP-9/NGAL was observed. Finally, MMP-9/NGAL and ferritin were found to be negatively correlated under pathological conditions. Scatter plot graphs of significant correlations have been presented in Appendix A (Appendix A).

## 4. Discussion

In the present retrospective observational study, we analyzed tissue gene expression of *LCN-2* and *MMP-9* and the circulating levels of LCN-2, MMP-9, and the MMP-9/NGAL complex in a selected cohort of patients with a first diagnosis of UADT carcinomas. The strict criteria limited patients to those who had never received any kind of treatment (interventional and/or pharmacological) before the recruitment, reducing the risk of potential confounding factors.

Tissue *LCN-2* and *MMP-9* gene analysis showed a decrease in *LCN-2* and an increase in *MMP-9* expression in tumoral tissues with respect to peri-tumoral portions. Interestingly, serum levels of LCN-2 were significantly lower than those of controls, whilst its conjugated form with MMP-9 (MMP-9/NGAL complex) was higher when compared to healthy people. As expected, MMP-9 circulating levels were elevated in cancer patients. Finally, ROC curve analysis disclosed an interesting potential value of serum LCN-2 to discriminate between healthy and unhealthy people, compared to the two other proteins analyzed.

UADT cancers represent a major global health challenge, with strong epidemiological differences related to behavioral risk factors, viral infections, and geographic variables [1,2]. A healthy lifestyle (reducing tobacco and alcohol consumption and adopting a healthy diet) and Human Papillomavirus vaccination are crucial and controllable preventive measures to reduce the incidence and improve the prognosis of these tumors [28]. Nevertheless, the identification of biomarkers for the diagnosis and/or progression monitoring of the disease or to evaluate the efficacy of a treatment is a growing field of interest in oncology.

LCN-2 has been studied as a biomarker in tumors due to its overexpression in various types of cancer, making it useful for diagnosis, prognosis, and treatment monitoring [18,31,32]. LCN-2 levels are elevated in several types of cancer, including breast cancer [33], colorectal cancer [34], and lung cancer [35]. Overexpression of LCN-2 has been associated with a worse prognosis in various tumors, particularly in association with metastatic invasion [36,37,38,39,40]. The presence of LCN-2 in plasma or tumor fluid may indicate a functional inflammation in response to tumor proliferation. In breast cancer, for example, LCN-2 has been found elevated in patients with advanced tumors and lymph node metastases [33,36].

Concerning UADT carcinomas, some research evidenced contrasting findings. Up-regulation of LCN-2 was found in oral squamous cell carcinoma (OSCC) tissues (tongue) and oral carcinoma cell lines in relation to cancer progression (higher levels in well-differentiated tissues with respect to poorly/moderately differentiated cells), suggesting that LCN-2 could be a useful marker of tumor-cell differentiation [41,42,43,44]. The only study that took into consideration circulating (plasmatic) levels of LCN-2, MMP-9, and the MMP-9/NGAL complex disclosed increased levels of the three molecules in patients with OSCC compared to those in healthy controls [42]. Both these studies presented results partially in contrast with ours (tissue and circulating LCN-2 increased levels). We hypothesize that these discrepancies may be due, at least in part, to the inclusion/exclusion criteria for patient enrollment. In the cited studies [41,42], no information was available on potential pharmacological and/or interventional treatments, at the time of sampling, that could significantly affect the results.

On the other hand, some research studies were in line with our findings. Monisha and collaborators [20] showed that, in tissue microarray slides for head and neck cancers and OSCC, LCN-2, detected in all samples analyzed, was weakly expressed compared to healthy tissues. Authors also demonstrated that *LCN-2* knockdown cells possessed higher invasive ability than control cells, as shown in in vitro invasion and migration assays. Moreover, in the same cells, MMP-9 was found to be overexpressed (in line with our findings) and presumably responsible for the increase in cell motility [20].

In another study [45], Lin et al. demonstrated that LCN-2 expression decreased in patients with OSCC with lymph node metastasis compared with that in patients without metastasis. A higher LCN-2 expression correlated with the survival of patients with OSCC. Furthermore, LCN-2 overexpression in OSCC cells reduced in vitro migration and invasion and in vivo metastasis, whereas its silencing induced an increase in cell motility [45]. This effect was also confirmed in nasopharyngeal carcinoma cell lines, where downregulation of LCN-2 levels via siRNA targeting *LCN-2* enhanced cell migration and invasion [46].

The intriguing observation that both upregulation and downregulation of LCN-2 have been associated with a malignant phenotype in different cancer types powerfully suggests distinctive transduction pathways that converge on an overall pro-carcinogenic role of the LCN-2 system. In support of this hypothesis, our study demonstrated a dual trend of the LCN-2 system in the serum of UADT cancer patients. Whilst the free form of LCN-2 decreased when compared to healthy controls, the form complexed with MMP-9 (MMP-9/NGAL complex) strongly increased in oncological patients, as well as the unconjugated MMP-9. Moreover, tissue analysis carried out through droplet digital PCR, technology that provides ultrasensitive nucleic acid detection and absolute quantification, disclosed that both *LCN-2* and *MMP-9* gene expression levels were comparable to that of the circulating levels.

The LCN-2 reduction in serum and tissue samples of oncological patients was in line with its role in promoting invasion and migration in UADT carcinomas [47], as well as in colorectal cancer and breast cancer when upregulated [32,34]. Moreover, as demonstrated by Monisha’s study, the knockdown of *LCN-2* in tumoral cells activated mTOR signaling via the LKB1-AMPK-p53-Redd1 pathway, inducing resistance to autophagy and providing a survival advantage to OSCC tumoral cells [20].

Regarding the mechanism of action of LCN-2 in its conjugated form with MMP-9 (MMP-9/NGAL complex), overexpressed in the serum of patients with UADT cancer, the role of the two proteins in the dynamics of metastasis and tumor progression comes into play. MMP-9 is a metalloproteinase present in the extracellular matrix capable of degrading some components of the matrix, such as type IV collagen, gelatin, and other structural proteins [47,48]. MMP-9 is also expressed, in addition to neutrophils, by other cells, such as tumor cells. In this context, MMP-9 plays a critical role in tumor invasion and metastasis by facilitating, through the degradation of the extracellular matrix, infiltration, and diffusion into surrounding tissues [48,49,50]. NGAL and MMP-9 are also associated with the establishment of an inflammatory response, a favorable factor for tumor progression [50]. It has been shown that NGAL protein binds to MMP-9, forming the MMP-9/NGAL complex that improves the stability of the enzyme, promoting its extracellular matrix remodeling activity and ultimately preparing the tissue for the spread of tumor cells [51,52]. A recent study demonstrated that the MMP-9/NGAL complex associated with lysyl oxidase-like 2 (LOXL2) facilitated tumor metastasis in esophageal cancer [51].

Another interesting finding observed in our study derived from Spearman’s correlation analysis of serum values and, in particular, the relationship between LCN-2 (free and conjugated forms) and the circulating proteins (ferritin and transferrin) involved in iron trafficking. Iron, although present in modest quantities, plays an important role in the organism [53,54]. Maintaining the balance of its levels is of fundamental importance since both a decrease and an increase can be the cause of numerous pathological conditions, including the promotion of tumor initiation, growth, and metastasis [55,56,57]. LCN-2 plays an important role in iron binding and sequestration [58,59]. This process helps to limit the availability of iron to pathogens, which are dependent on iron for their growth and survival [60]. The formation of the complex with iron promotes its transport to the appropriate sites, such as immune system cells that require it for their functioning [54]. Moreover, LCN-2, by limiting the availability of free iron, reduces the possibility of triggering lipid peroxidation, a process capable of causing cellular and tissue damage, especially in tissues involved in chronic inflammatory processes [61,62]. These findings are consistent with what was observed in our study. LCN-2 levels in healthy individuals are negatively correlated with ferritin (a protein that mainly promotes iron accumulation in specific reserve tissues) and positively with transferrin (which transports iron to cells that need it). In UADT cancer patients, the drastic reduction in LCN-2/ferritin correlation is followed by the loss of relationship with transferrin, supporting the idea that the modification in LCN-2 levels also impacts the processes involved in the endogenous regulation of circulating iron. The strong reduction of serum LCN-2 levels in cancer patients could contribute to making available circulating levels of iron useful to tumor cells for their active proliferation. The concomitant increase in ferritin observed could contribute to the accumulation of intracellular iron used for tumor growth.

## 5. Conclusions and Perspectives

The limited number of samples analyzed may represent a weakness of our study, although the stringent selection of patients only with the first diagnosis of UADT carcinoma constitutes a solid starting point for further investigations. Our study confirms and extends the knowledge on the involvement of the LCN-2 system in upper aerodigestive tract tumors, revealing a potentially different role of the free form (LCN-2) and the one conjugated with MMP-9 (MMP-9/NGAL complex) in this pathological condition. These findings stimulate further investigations on the potential value of the two forms as markers of different stages in the carcinogenesis process. On the one hand, the conjugated form of LCN-2 could be an index of tumor invasiveness, whilst the free form could be of support in monitoring therapeutic efficacy or detecting tumor recurrence. In line with this consideration, ROC curve analysis for LCN-2, showing the significant ability to discriminate between healthy controls and cancer patients, offers an interesting food for thought in this direction. Understanding these mechanisms would offer potential innovative therapeutic approaches for the treatment of cancer.

## Figures and Tables

**Figure 1 cells-14-00506-f001:**
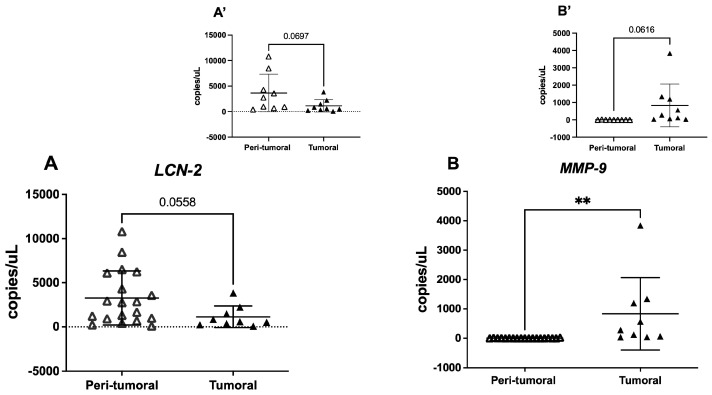
*LCN-2* and *MMP-9* gene expression in the available peri-tumoral and tumoral tissues in cancer patients. The figure represents the mean ± SD of *LCN-2* and *MMP-9* expression, measured in RNA copy number/μL, as determined by droplet PCR. Panels (**A**,**B**) compare total peri-tumoral vs. tumoral tissues, while panels (**A’**,**B’**) present data from matched patient samples. Student’s *t*-test ** *p* < 0.01.

**Figure 2 cells-14-00506-f002:**
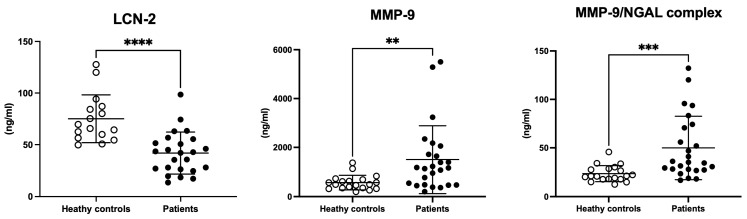
LCN-2, MMP-9, and MMP-9/NGAL complex circulating levels in healthy controls vs. UADT cancer patients. Scatter dot plots represent the mean ± SD of serum levels from controls and patients. Student’s *t*-test ** *p* < 0.01; *** *p* < 0.001; **** *p* < 0.0001.

**Figure 3 cells-14-00506-f003:**
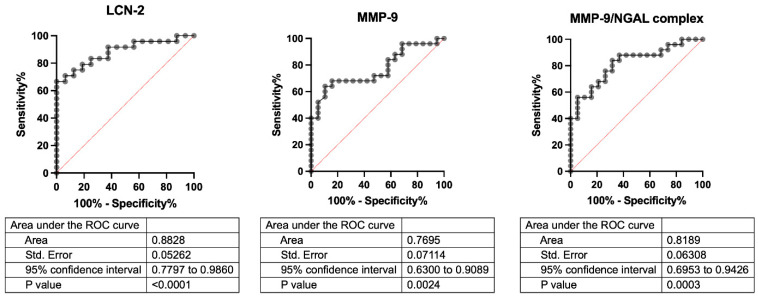
ROC curves for circulating levels of LCN-2, MMP-9, and MMP-9/NGAL complex in healthy controls vs. UADT cancer patients. Below each graph, the following have been shown: area under the curve, standard error, 90% confidence interval, and *p* values.

**Table 1 cells-14-00506-t001:** General characteristics of the enrolled patients: gender, age, cancer district, clinical (c) or pathological radiation (TNM) (p), availability for serum and tissue sampling.

Patient ID#	Sex	Age (Years)	Cancer District	p/cTNM	Serum Sampling (Yes/No)	Tissue Sampling (Yes/No)	
						Peri- Tumoral	Tumoral
1	M	60	Hypopharynx	cT4bN3bM0	Yes	No	No
2	F	57	Hypopharynx	pT4aN0M0	Yes	No	No
3	M	72	Larynx	pT4aN0M0	Yes	No	No
4	M	58	Hypopharynx	pT3N3bM0	Yes	No	No
5	M	54	Larynx	pT3N3bM0	Yes	No	No
6	M	63	Larynx	pT4aN0M0	Yes	Yes	No
7	M	57	Tongue	pT3N1M0	Yes	Yes	No
8	M	49	Larynx	pT3N0M0	Yes	Yes	No
9	M	66	Oropharynx	cT4N2bM0	Yes	No	No
10	F	58	Tongue	pT3N3bM0	Yes	Yes	No
11	M	59	Upper Gingiva	cT4bN2cM1	Yes	No	No
12	M	60	Tongue	cT4aN2bM0	Yes	No	No
13	M	53	Oropharynx	cT4aN3bM0	Yes	No	No
14	M	46	Tongue	pT1N0M0	Yes	No	No
15	M	65	Larynx	pT3N0M0	Yes	No	No
16	M	70	Larynx	pT4aN3bM0	Yes	Yes	No
17	F	75	Larynx	pT3N0M0	Yes	Yes	No
18	M	56	Larynx	pT3N0M0	Yes	Yes	No
19	M	71	Larynx	pT3N3bM0	Yes	Yes	No
20	F	60	Larynx	pT2N0M0	Yes	Yes	No
21	M	67	Larynx	pT4aN0M0	Yes	Yes	No
22	M	78	Larynx	pT3N0M0	Yes	Yes	Yes
23	M	56	Larynx	pT4aN1M0	Yes	Yes	Yes
24	F	73	Larynx	pT3N3bM0	Yes	Yes	Yes
25	M	77	Larynx	pT3N0M0	Yes	Yes	Yes
26	F	62	Larynx	pT2N0M0	No	Yes	Yes
27	M	69	Larynx	pT4aN0M0	No	Yes	Yes
28	M	76	Larynx	pT2N0M0	No	Yes	Yes
29	M	65	Larynx	pT4aN2bM0	No	Yes	Yes
30	M	60	Larynx	pT4aN3bM0	No	Yes	Yes

**Table 2 cells-14-00506-t002:** Hematologic parameters in healthy controls and UADT cancer patients.

Parameter (Range)	Sex	Healthy Controls (n. 20; M16/F4)	Cancer Patients (n. 25; M20/F5)	*p*-Value	(df), F
Glucose (70.3–100.9 mg/dL)	tot	88.31 ± 2.23	113.32 ± 6.46	0.002	(1,42), 10.591
	M	87.6 ± 2.51	111 ± 19.4	*p*_(disease)_ 0.007 *p*_(gender)_ 0.464	(1,40) F_(disease)_ 7.99 F_(gender)_ 0.646
	F	91 ± 5.35	122 ± 6.71		
UREA (10.20–49.80 mg/dL)	tot	16.94 ± 1.15	31.8 ± 3.51	<0.001	(1,42), 12.749
	M	17.45 ± 1.29	30.47 ± 3.85	*p*_(disease)_ 0.002 *p*_(gender)_ 0.686	(1,40) F_(disease)_ 11.41 F_(gender)_ 0.16
	F	15 ± 2.64	37.18 ± 8.92		
Phosphate (2.80–4.60 mg/dL)	tot	3.12 ± 0.10	3.52 ± 0.10	0.01	(1,42), 7.364
	M	2.98 ± 0.09	3.46 ± 0.103	*p*_(disease)_ 0.081 *p*_(gender)_ 0.006	(1,40) F_(disease)_ 3.215 F_(gender)_ 8.468
	F	3.65 ± 0.17	3.78 ± 0.32		
Protein (60–82 g/L)	tot	72.05 ± 0.83	66.6 ± 1.58	0.007	(1,42), 7.911
	M	71.2 ± 0.87	66.35 ± 1.86	*p*_(disease)_ 0.013 *p*_(gender)_ 0.301	(1,40) F_(disease)_ 6.806 F_(gender)_ 1.097
	F	72.25 ± 1.55	67.4 ± 2.89		
Albumin (35–55 g/L)	tot	44.16 ± 0.57	40.64 ± 1.19	0.03	(1,42), 5.072
	M	43.9 ± 0.71	40.1 ± 1.44	*p*_(disease)_ 0.130 *p*_(gender)_ 0.340	(1,40) F_(disease)_ 2.393 F_(gender)_ 0.933
	F	45 ± 0.41	42.8 ± 3.0		
Bilirubin–Direct (<0.20 mg/dL)	tot	0.15 ± 0.02	0.31 ± 0.04	0.002	(1,42), 10.481
	M	0.15 ± 0.03	0.32 ± 0.05	*p*_(disease)_ 0.013 *p*_(gender)_ 0.832	(1,40) F_(disease)_ 6.729 F_(gender)_ 0.05
	F	0.13 ± 0.02	0.31 ± 0.05		
Iron (64.8–174.90 ug/dL)	tot	97.89 ± 5.84	75.14 ± 9.17	0.059	(1,42), 3.775
	M	99.73 ± 7.24	78.28 ± 10.70	*p*_(disease)_ 0.09 *p*_(gender)_ 0.410	(1,40) F_(disease)_ 2.883 F_(gender)_ 0.694
	F	91 ± 6.10	78.28 ± 10.70		
PCR (100–6000 ug/L)	tot	1392 ± 319	29212± 11387	0.04	(1,42), 4.372
	M	1673 ± 274	32955 ± 13990	*p*_(disease)_ 0.197 *p*_(gender)_ 0.587	(1,40) F_(disease)_ 1.721 F_(gender)_ 0.300
	F	2350 ± 1211	14240 ± 10081		
Transferrin (2.15–3.65 g/L)	tot	2.76 ± 0.10	2.23 ± 0.11	0.001	(1,42), 2.999
	M	2.70 ± 0.11	2.24 ± 0.13	*p*_(disease)_ 0.002 *p*_(gender)_ 0.617	(1,40) F_(disease)_ 10.808 F_(gender)_ 0.254
	F	2.97 ± 0.19	2.17 ± 0.21		
Creatinine (M: 0.70–1.20 mg/Dl F: 0.50–0.9 mg/dL)	M	1.04 ± 0.02	0.85 ± 0.06	0.012	(1,33), 7.112
	F	0.78 ± 0.05	0.64 ± 0.08	0.189	(1,7), 2.117
gGT (M: 10.00–40.00 U/L F: 7.00–35.00 U/L)	M	17.88 ± 1.54	55 ± 18.24	0.01	(1,33), 3.073
	F	16.25 ± 4.25	26.6 ± 3.72	0.109	(1,7), 3.374
LDH (M: 135.00–225.00 U/L F: 135.00–214.00 U/L)	M	161.26 ± 8.14	129.95 ± 9.28	0.02	(1,33), 5.950
	F	173.5 ± 12.63	138 ± 28.53	0.334	(1,7), 1.077
CPK (M: 20–200 U/L F:20–180 U/L)	M	139.27 ± 27.68	51.90 ± 12.91	0.004	(1,33), 9.627
	F	89.75 ± 13.70	30.20 ± 5.40	0.003	(1,7), 19.454
Ferritin (M: 30–400 ug/L F: 15–150 ug/L)	M	67 ± 10.89	402.95 ± 87.45	0.002	(1,33), 10.891
	F	61.50 ± 18.40	313 ± 76.35	0.02	(1,7), 8.154

**Table 3 cells-14-00506-t003:** Spearman’s correlation between LCN-2, MMP-9, and NGAL/MMP-9 complex vs. hematologic parameters, in healthy controls and UADT cancer patients.

Spearman’s Correlations in Healthy Controls						
Variable		LCN-2	MMP-9/NGAL	MMP-9	Transferrin	Ferritin
LCN-2	Spearman’s rho	—				
	*p*-value	—				
MMP-9/NGAL	Spearman’s rho	0.236	—			
	*p*-value	0.397	—			
MMP-9	Spearman’s rho	−0.443	0.751 ***	—		
	*p*-value	0.100	<0.001	—		
Transferrin	Spearman’s rho	0.524 *	0.203	−0.106	—	
	*p*-value	0.040	0.418	0.674	—	
Ferritin	Spearman’s rho	−0.680 **	−0.338	−0.137	−0.688 **	—
	*p*-value	0.004	0.171	0.587	0.001	—
**Spearman’s Correlations in UADT Cancer Patients**						
**Variable**		**LCN-2**	**MMP-9/NGAL**	**MMP-9**	**Transferrin**	**Ferritin**
LCN-2	Spearman’s rho	—				
	*p*-value	—				
MMP-9/NGAL	Spearman’s rho	0.598 **	—			
	*p*-value	0.002	—			
MMP-9	Spearman’s rho	0.242	0.674 ***	—		
	*p*-value	0.242	<0.001	—		
Transferrin	Spearman’s rho	0.155	0.130	0.053	—	
	*p*-value	0.459	0.534	0.800	—	
Ferritin	Spearman’s rho	−0.400 *	−0.479 *	−0.254	−0.425 *	—
	*p*-value	0.049	0.016	0.220	0.034	—

* *p* < 0.05, ** *p* < 0.01, *** *p* < 0.001.

## Data Availability

Data are available on request due to ethical reasons.

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
