# Peer review of "Lipocalin-2, Matrix Metalloproteinase-9, and MMP-9/NGAL Complex in Upper Aerodigestive Tract Carcinomas: A Pilot Study"

_cells, 2025, doi:10.3390/cells14070506_

Round 1

Reviewer 1 Report

Comments and Suggestions for Authors

The manuscript by Cavalcanti L. et all  presents an innovative approach to understanding upper aerodigestive tract carcinomas by exploring the roles of Lipocalin-2 and Matrix Metalloprotease 9 as dual biomarkers and their complex interactions in cancer progression. The manuscript extends its analysis to circulating levels of these biomarkers, potentially offering a less invasive diagnostic approach for UADT cancers. The study connects these biomarkers to key aspects of cancer progression and metastasis, which is crucial in understanding how UADT carcinomas spread.

Improvement suggestions:

1) The sample size for the cancer group (30 participants) and healthy controls (20 participants) is relatively small, and there is no mention of a power analysis to justify this sample size.

2) The healthy control group is age- and sex-matched; are there other confounding factors that could influence the results?

3) The study uses tissue samples from 9 tumor and 19 non-tumor tissues; are any of the non-tumor tissues matching tumor tissues? what does the difference come from?

4) The investigated cohort consists mainly of men in both cancer patients and controls, which may limit the usefulness of the findings. 

5) This study would benefit from analysis how LCN-2 and MMP-9 expression correlate with tumor stage, tumor grade and prognosis in UADT patients.

Author Response

The manuscript by Cavalcanti L. et all  presents an innovative approach to understanding upper aerodigestive tract carcinomas by exploring the roles of Lipocalin-2 and Matrix Metalloprotease 9 as dual biomarkers and their complex interactions in cancer progression. The manuscript extends its analysis to circulating levels of these biomarkers, potentially offering a less invasive diagnostic approach for UADT cancers. The study connects these biomarkers to key aspects of cancer progression and metastasis, which is crucial in understanding how UADT carcinomas spread.

Improvement suggestions:

  • The sample size for the cancer group (30 participants) and healthy controls (20 participants) is relatively small, and there is no mention of a power analysis to justify this sample size.

Reply: we thank the review for this comment. Our study is designed as a pilot study. For the calculation of the sample size in a pilot study, we relied on Power Analysis, setting acceptable conditions of a 5% Type I error and a test power of 90% (beta error= 10%). The tests were conducted on independent groups using a two-tailed approach to increase the reliability of the test itself. To reduce sample variance, the two populations (cases and controls) were matched by sex, age, and other potentially confounding factors (such as smoking and alcohol consumption). Regarding the "case" population, as described in material and methods section (page 3 lines 100-109) we recruited adult male and female patients entering the hospital for the first diagnosis of UADT carcinomas. In terms of the exclusion criteria, we did not include patients with presence of cancer relapse, under drug treatments (such as chemotherapies, anti-inflammatory, and immunosuppressants), radiotherapy and/or immunotherapy; patients suffering from severe infectious diseases (HIV, HBV, and HCV), or other ongoing inflammatory, cardiovascular, endocrine, and autoimmune disorders.

As the starting variable for the power analysis, we selected the serum levels of LCN-2, as we expected the smallest difference between the diseased and healthy groups—i.e., the smallest delta—given the limited availability of scientific studies on the topic.

According to previous studies, we expected to observe an LCN-2 concentration of approximately 45 ng/mL in patients, with a standard deviation of about 30 ng/mL, whereas healthy individuals typically present a concentration of around 77 ng/mL, with a standard deviation of about 20 ng/mL.

Below is the calculation report from G*Power:

t tests - Means: Difference between two independent means (two groups)

Analysis:           A priori: Compute required sample size

Input:

Tail(s)                      

Two

Effect size d

1,255143

α err prob

0,05

Power (1-β err prob)

0,9

Allocation ratio N2/N1

1

Output:

Noncentrality parameter δ

3,4373507

Critical t                  

2,0484071

Df

28

Sample size group 1        

15

Sample size group 2        

15

Total sample size            

30

Actual power                  

0,9126571

This allowed us to be confident that with at least 15 samples per group, we would be able to detect real differences—if they existed and were not due to chance—between the two groups regarding the biomarker under study.

Moreover, the following graph allowed us to ensure greater test power with the addition of just a few more samples, which indeed occurred during the recruitment of both patients and healthy individuals. This increase benefited not only the calculated variable but also the other monitored markers.

We have modified the title of the manuscript in the following: Lipocalin-2, Matrix Metalloproteinase-9 and MMP-9/NGAL complex in Upper Aerodigestive Tract Carcinomas: a pilot study.

A sentence has been added in the Participants’ Selection and Sampling section (pages 2-3 lines 94-99). A more extensive explanation has been reported in Supplementary materials.

  • The healthy control group is age- and sex-matched; are there other confounding factors that could influence the results?

Reply: we thank the review for this comment. We matched cases and controls not only by age and sex but also based on tobacco and alcohol consumption (according to interviews conducted during recruitment), considering these factors as variables present in UADTS patients. A clarifying sentence has been added (page 3 lines 107-109).

  • The study uses tissue samples from 9 tumor and 19 non-tumor tissues; are any of the non-tumor tissues matching tumor tissues? what does the difference come from?

Reply: we thank the review for this comment. the disparity in the tumor and non-tumor sample size was due to availability during the sampling phase. We have integrated the figure 1, the legend and the results by incorporating data from peri-tumoral and tumoral tissues collected from the same UADT cancer patients (n=9) (page 8, lines 240-249).

  • The investigated cohort consists mainly of men in both cancer patients and controls, which may limit the usefulness of the findings. 

Reply: Epidemiological data have clearly demonstrated that, globally, UADT cancers are more common in men than in women, with a male-to-female ratio of approximately 2:1, and in adults over 50 years of age (doi:10.3390/medsci11020042; doi.org/10.3322/caac.21660). These data may certainly vary across different geographical areas of the world, depending on various factors that can influence cancer risk (such as smoking, alcohol consumption, HPV, etc.).

Our strict inclusion criteria, which limited the analysis to only patients with a first diagnosis of UADT cancer and without prior pharmacological and/or interventional treatments, have undoubtedly contributed to exacerbating the already significant gender disparity (1:4 in our pilot study). Therefore, we believe that the research findings in our study deserve consideration at a global level.

A clarifying sentence and reference have been added (page 5, lines 193-195).

  • This study would benefit from analysis how LCN-2 and MMP-9 expression correlate with tumor stage, tumor grade and prognosis in UADT patients.

Reply: we thank the review for this comment. We completely agree with the referee on the benefit derived from the correlation analysis between LCN-2 and MMP-9 expression and tumor stage, tumor grade and prognosis in our group of patients.

Concerning this point, as indicated in the discussion (pages 11, lines 326-338) there is evidence that LCN-2 expression on tissue microarrays of patients with OSCC with lymph node metastasis decreased, compared with that in patients without metastasis, and that a higher LCN-2 expression correlated with the survival of patients with OSCC (doi:10.3390/cancers10070228; doi.org/10.1093/carcin/bgw050).

The limited number of tumoral tissues in this pilot study does not allow for such considerations at this stage. Nevertheless, this falls outside the scope of our work, which aims to investigate the relationship between serum findings and tissue expression through the analysis of LCN-2 and MMP-9 gene expression.

Reviewer 2 Report

Comments and Suggestions for Authors

This study investigates the roles of Lipocalin-2 (LCN-2), Matrix Metalloproteinase-9 (MMP-9), and the MMP-9/NGAL complex in upper aerodigestive tract carcinomas. The findings suggest their potential as biomarkers for cancer progression and prognosis, emphasizing the need for further clinical validation.

However, several concerns need to be addressed:

  1. Figure 1: The legend describes comparisons between healthy and tumor samples, but the figure itself appears to compare peritumoral and tumor tissues. Additionally, the sample count (healthy: 19, tumor: 9) is unclear—where are the remaining samples? Are they from the same patients? The figure should explicitly indicate the expression levels of each sample to provide clarity.
  2. Figure 2: The sample size is inconsistent. The figure mentions 20 controls and 25 patients, whereas the characteristics table lists 30 patients. Clarification is required.
  3. Figure 3: The selection of samples is unclear. Since different TNM stages represent distinct disease states, mixing T1, T2, T3, and T4 data may lead to misleading conclusions. Please specify which samples were used and consider stratifying the data by stage.
  4. Abstract: It should explicitly address: The diagnostic potential of LCN-2 vs. the MMP-9/NGAL complex.Whether these biomarkers can distinguish between cancer stages. The clinical relevance of these findings for early detection or treatment response monitoring.
  5. Gene Nomenclature: Ensure that gene and protein names follow the Gene/Protein Nomenclature Guidelines to maintain accuracy and consistency.
  6. Statistical analysis methods should be mentioned in the figures.
Comments on the Quality of English Language

The English could be improved to more clearly express the research.

Author Response

This study investigates the roles of Lipocalin-2 (LCN-2), Matrix Metalloproteinase-9 (MMP-9), and the MMP-9/NGAL complex in upper aerodigestive tract carcinomas. The findings suggest their potential as biomarkers for cancer progression and prognosis, emphasizing the need for further clinical validation.

However, several concerns need to be addressed:

  1. Figure 1: The legend describes comparisons between healthy and tumor samples, but the figure itself appears to compare peritumoral and tumor tissues. Additionally, the sample count (healthy: 19, tumor: 9) is unclear—where are the remaining samples? Are they from the same patients? The figure should explicitly indicate the expression levels of each sample to provide clarity.

Reply: we thank the review for this comment. The disparity in the tumor and non-tumor sample size was due to availability during the sampling phase. We have integrated the figure, the legend and results by incorporating data from peri-tumoral and tumoral tissues collected from the same UADT cancer patients (n=9) (page 8, lines 240-249).

  1. Figure 2: The sample size is inconsistent. The figure mentions 20 controls and 25 patients, whereas the characteristics table lists 30 patients. Clarification is required.

Reply:  In Table 1 (now modified in its presentation as per the request of another reviewer, page 5, lines 199-200) we have indicated the total number of recruited patients (30) and controls (20), as well as how many were eligible for serum sampling (25 and 20 respectively) and tissue collection (9 and 19 respectively).

Figure 2 has now been modified to include a scatter dot plot graph (page 9)

  1. Figure 3: The selection of samples is unclear. Since different TNM stages represent distinct disease states, mixing T1, T2, T3, and T4 data may lead to misleading conclusions. Please specify which samples were used and consider stratifying the data by stage.

Reply: we thank the referee for the comment. The sample selection in Figure 3 (ROC curves) includes all control and UADT cancer patients. We agree with the referee that TNM stages represent distinct disease states and may provide different information. However, in our pilot study, a differential analysis considering subgrouping based on T stage (as well as N involvement) did not reveal any significant differences in any of the biomarkers analyzed (we have added these results as supplementary results and a sentence in page 9, lines 259-261).

Because of the contrasting data reported in the literature on this topic (largely discussed in the proper section, see pages 11-12 lines 326-352) our first aim was to give a contribution in defining the involvement and modulation of LCN-2 system in UADTs cancer, selecting patients at the first diagnosis and avoid of any pharmacological/interventional treatment.  We unrevealed not only the unusual decrease of the free form of LCN-2 (compared to what happened in other oncological cases), but also the strong increase of the conjugated complex with MMP-9, potentially disclose different mechanism underlying the cancerogenic process.

ROC analysis showed that the free form of LCN-2 outperformed the other biomarkers in distinguishing healthy individuals from those with the disease, highlighting its potential clinical value for disease stage monitoring.

In our opinion, these findings lay the foundation for future investigations into the system's potential as a biomarker for remission stages, response to therapies, and more, as discussed in the section.

  1. Abstract: It should explicitly address: The diagnostic potential of LCN-2 vs. the MMP-9/NGAL complex.Whether these biomarkers can distinguish between cancer stages. The clinical relevance of these findings for early detection or treatment response monitoring.

Reply: according to the referee, abstract has been revised

  1. Gene Nomenclature: Ensure that gene and protein names follow the Gene/Protein Nomenclature Guidelines to maintain accuracy and consistency.

Reply: according to http://www.genenames.org/guidelines.html we have corrected the errors.

  1. Statistical analysis methods should be mentioned in the figures.

Reply: according to referee suggestion we legends have been updated.

Reviewer 3 Report

Comments and Suggestions for Authors

This manuscript analyzes the serum levels and tissue levels of lipocalin-2 (lcn-2), matrix metalloproteinase-9 (MMP-9), and MMP-9/NGAL 2 complex in the upper aerodigestive tract (UADT) carcinomas.

The cohort of 30 patients (7 tumor tissues and 19 healthy tissues) is characterized by the absence of any treatment. 

The authors reported some differences between the healthy donors and patients have been reported.

The experiments are well performed and presented. The analysis is in enough detail and the presentation is overall good.

It remains to define whether it is worth publishing the data shown or not.

I strongly suggest inserting some additional data and performing new experiments as well.

1- the histological analysis of specimens for the different markers in the tumor and healthy specimens should be performed to determine where (in which cells by IHC and appropriate software analysis and quantification) the lcn-2, MMP-9, and the  MMP-9/NGAL 2 complex are expressed.

2- the comparison of serum levels should be performed considering simply inflammatory conditions to define if the differences reported are related to inflammation or tumor development.

3- results obtained by these two approaches should be discussed with what is already present in the literature.

4- the functional significance of what is reported is not clear. Also, it is not so clear whether the amount detected is functional.

5- the data for each patient or healthy donor should be plotted, beside the mean and SEM (it would be better to have the SD instead).

Looking at the discussion it appears that some of the biomarkers found are present in inflammatory conditions. Also, it is well-known that tumors can be considered, at least by some authors, as a chronic inflammatory disease. Overall, this reviewer does not take a strong message from the manuscript in this form because the data reported remains mainly descriptive and typical of the cohort analyzed. The selection of untreated patients is of interest but it is not reported what can happen after the treatment to understand if what is analyzed is worth to be analyzed to monitor the disease, the residual disease, the clinical outcome, the development of resistance to therapy, and so on.

The experiments suggested can give some insight but this is not certain, of course. At least, they can give a more comprehensive scenario. Several others can be performed to give a better description of the variability of  MMP-9/NGAL 2 complex in UADT carcinomas (analysis at the single cell level and so on) but it is questionable they will clarify something more. However, the simple description could allow the generation of new ideas on the topic.

Author Response

This manuscript analyzes the serum levels and tissue levels of lipocalin-2 (lcn-2), matrix metalloproteinase-9 (MMP-9), and MMP-9/NGAL 2 complex in the upper aerodigestive tract (UADT) carcinomas.

The cohort of 30 patients (7 tumor tissues and 19 healthy tissues) is characterized by the absence of any treatment. 

The authors reported some differences between the healthy donors and patients have been reported.

The experiments are well performed and presented. The analysis is in enough detail and the presentation is overall good.

It remains to define whether it is worth publishing the data shown or not.

I strongly suggest inserting some additional data and performing new experiments as well.

  • the histological analysis of specimens for the different markers in the tumor and healthy specimens should be performed to determine where (in which cells by IHC and appropriate software analysis and quantification) the lcn-2, MMP-9, and the  MMP-9/NGAL 2 complex are expressed.

Reply: we do agree with the reviewer that histological analysis to determine in which cells lcn-2, MMP-9, and the MMP-9/NGALcomplex are expressed could add interesting information on this topic, different and complementary to what we have done in our experimental plan. Nevertheless, we provided strong information in whole tissue extract, by using digital droplet PCR, which provides extremely accurate quantification of DNA at high levels of sensitivity and specificity, without the need for standard assays [doi: 10.1002/0471142905.hg0724s82).

This approach aligns more closely with the primary aim of our study, which was to analyze the expression levels (serum and/or tissue) of the LCN-2 system (both free and complexed forms) rather than to define its cellular localization.

  • the comparison of serum levels should be performed considering simply inflammatory conditions to define if the differences reported are related to inflammation or tumor development.

Reply: Our pilot study highlighted that UADT cancer patients showed lower serum LCN-2 levels compared to healthy controls, despite significantly higher levels of C-reactive protein, a marker of inflammation. To our knowledge, inflammatory conditions associated with various pathological conditions have predominantly been linked to increased serum/plasma LCN-2 levels (PMID: 26116587; PMID: 23242471; PMID: 28127551).

Regarding cancer pathology, in several studies, serum LCN-2 levels have been compared to those of healthy donors, primarily for the initial screening of its potential as a biomarker, independently of the presence of inflammatory conditions or other potential interacting factors, considering cancer as a multifactorial disease (PMID: 30829613; PMID: 39973816; PMID: 23389669; PMID: 23838135; PMCID: PMC2775750). These considerations can be extended to all the markers analyzed in our study.

  • results obtained by these two approaches should be discussed with what is already present in the literature.

Reply: see the reply to point 1 and 2

  • the functional significance of what is reported is not clear. Also, it is not so clear whether the amount detected is functional.

Reply: Because of the contrasting data reported in the literature on this topic (extensively discussed in the appropriate section, see page 11-12, lines 326-352) our first aim was to give our contribution in defining the involvement and modulation of LCN-2 system in UADTs cancer, selecting patients at the first diagnosis and avoid of any pharmacological/interventional treatment. We unrevealed not only the unusual decrease of the free form of LCN-2 (compared to other oncological conditions), but also the strong increase of the conjugated complex with MMP-9, potentially disclose different mechanism underlying the cancerogenic process. Moreover, ROC analysis showed that the free form of LCN-2 outperformed the other biomarkers in distinguishing healthy individuals from those with the disease, highlighting its potential clinical value for disease stage monitoring.

  • the data for each patient or healthy donor should be plotted, beside the mean and SEM (it would be better to have the SD instead).

Reply: as suggested by the referee all the graphs have been changed.

  • Looking at the discussion it appears that some of the biomarkers found are present in inflammatory conditions. Also, it is well-known that tumors can be considered, at least by some authors, as a chronic inflammatory disease. Overall, this reviewer does not take a strong message from the manuscript in this form because the data reported remains mainly descriptive and typical of the cohort analyzed.

Reply: see question 2

  • The selection of untreated patients is of interest but it is not reported what can happen after the treatment to understand if what is analyzed is worth to be analyzed to monitor the disease, the residual disease, the clinical outcome, the development of resistance to therapy, and so on.

Reply: We thank the referee for the comment. As discussed in point 4, due to the conflicting data reported in the literature on this topic (extensively addressed in the relevant section, see page 11-12, lines 326-352), our primary goal was to contribute to defining the involvement and modulation of the LCN-2 system in UADT cancer. To achieve this, we selected patients at first diagnosis, ensuring they had not undergone any pharmacological or interventional treatment.

Understanding what happens after treatment is crucial for determining whether the analyzed factors are useful for monitoring disease progression, residual disease, clinical outcomes, therapy resistance, and more. This could certainly be the focus of future studies.

  • The experiments suggested can give some insight but this is not certain, of course. At least, they can give a more comprehensive scenario. Several others can be performed to give a better description of the variability of  MMP-9/NGAL 2 complex in UADT carcinomas (analysis at the single cell level and so on) but it is questionable they will clarify something more. However, the simple description could allow the generation of new ideas on the topic.

Reply: see previous answers.

Reviewer 4 Report

Comments and Suggestions for Authors

Hello, 

Please find my suggestions attached!

Author Response

Dear Authors

Manuscript describes and concludes that Lipocalin-2, Matrix Metalloproteinase-9 tissue gene expression and LCN-2, MMP-9 and MMP-9/NGAL complex serum levels in patients with first diagnosis of Upper Aerodigestive Tract Carcinomas (UADT) cancer and avoid of any kind of treatment (pharmacological and/or interventional) and correlated the results with the observed hematological profile.

The following steps should provide more clear information for readers to enjoy it

  • Add up-to-date references in the introduction and discussion section.

Reply: According to the reviewer's comments, references have been updated where possible

  • 1. Participants’ Selection and Sampling – Small sample is size.

Reply: Our study is designed as a pilot study. For the calculation of the sample size in a pilot study, we relied on Power Analysis, setting acceptable conditions of a 5% Type I error and a test power of 90% (beta error= 10%). The tests were conducted on independent groups using a two-tailed approach to increase the reliability of the test itself. To reduce sample variance, the two populations (cases and controls) were matched by sex, age, and other potentially confounding factors (such as smoking and alcohol consumption). Regarding the "case" population, as described in material and methods section (page 3 lines 100-109), we recruited adult male and female patients entering the hospital for the first diagnosis of UADT carcinomas. In terms of the exclusion criteria, we did not include patients with presence of cancer relapse, under drug treatments (such as chemotherapies, anti-inflammatory, and immunosuppressants), radiotherapy and/or immunotherapy; patients suffering from severe infectious diseases (HIV, HBV, and HCV), or other ongoing inflammatory, cardiovascular, endocrine, and autoimmune disorders.

As the starting variable for the power analysis, we selected the serum levels of LCN-2, as we expected the smallest difference between the diseased and healthy groups—i.e., the smallest delta—given the limited availability of scientific studies on the topic.

According to previous studies conducted in past years, we expected to observe an LCN-2 concentration of approximately 45 ng/mL in patients, with a standard deviation of about 30 ng/mL, whereas healthy individuals typically present a concentration of around 77 ng/mL, with a standard deviation of about 20 ng/mL.

Below is the calculation report from G*Power:

t tests - Means: Difference between two independent means (two groups)

Analysis:           A priori: Compute required sample size

Input:

Tail(s)                      

Two

Effect size d

1,255143

α err prob

0,05

Power (1-β err prob)

0,9

Allocation ratio N2/N1

1

Output:

Noncentrality parameter δ

3,4373507

Critical t                  

2,0484071

Df

28

Sample size group 1        

15

Sample size group 2        

15

Total sample size            

30

Actual power                  

0,9126571

This allowed us to be confident that with at least 15 samples per group, we would be able to detect real differences—if they existed and were not due to chance—between the two groups regarding the biomarker under study.

Moreover, the following graph allowed us to ensure greater test power with the addition of just a few more samples, which indeed occurred during the recruitment of both patients and healthy individuals. This increase benefited not only the calculated variable but also the other monitored markers.

We have modified the title of the manuscript in the following: Lipocalin-2, Matrix Metalloproteinase-9 and MMP-9/NGAL complex in Upper Aerodigestive Tract Carcinomas: a pilot study.

A sentence has been added in the Participants’ Selection and Sampling section (pages 2-3 lines 94-99). A more extensive explanation has been reported in Supplementary materials.

  • 3. Tissue lcn-2 and mmp-9 gene expression – to confirm these data please perform protein expression – western blot technique.

Reply: The presence of proteins (LCN-2, MMP-9, and the MMP-9/NGAL complex) was detected in the serum of healthy individuals and UADT cancer patients using the ELISA method. We chose to analyze gene expression levels in tissue samples, as we believe this approach provides complementary information to protein detection by offering insights into gene expression regulation. Interestingly, data from both tissue analysis and serum levels showed the same trend: a decrease in LCN-2 serum levels and reduced LCN-2 gene expression in tissue samples from UADT patients, along with an increase in MMP-9 serum levels and an upregulation of MMP-9 gene expression in the same patients.

Reviewer 5 Report

Comments and Suggestions for Authors

Dear Authors

Manuscript describes and concludes that Lipocalin-2, Matrix Metalloproteinase-9 tissue gene expression and LCN-2, MMP-9 and MMP-9/NGAL complex serum levels in patients with first diagnosis of Upper Aerodigestive Tract Carcinomas (UADT) cancer and avoid of any kind of treatment (pharmacological and/or interventional) and correlated the results with the observed hematological profile.

The following steps should provide more clear information for readers to enjoy it

1) Add up-to-date references in the introduction and discussion section.

2) 2.1. Participants’ Selection and Sampling – Small sample is size.

3) 3.3. Tissue lcn-2 and mmp-9 gene expression – to confirm these data please perform protein expression – western blot technique.

Author Response

Brief summary:

This study included patients with newly diagnosed upper aerodigestive tract (UADT) carcinomas,

with no prior pharmacological or interventional treatments. The study investigated LCN-2 and

MMP-9 gene expression in tumor tissues, along with circulating levels of LCN-2, MMP-9, and the

MMP-9/NGAL complex in patient serum. The results showed downregulated LCN-2 and

upregulated MMP-9 gene expression in tumor tissues compared to healthy adjacent tissues.

Similarly, circulating LCN-2 levels were decreased, while MMP-9 levels were increased in cancer

patients compared to healthy controls. Notably, MMP-9/NGAL complex levels were significantly

elevated in the serum of UADT cancer patients.

Overall review:

The inclusion and exclusion criteria used by the authors to select the patient population are welldefined and appropriate. The decision to include only patients who had not undergone any prior

therapies is particularly valuable, as it helps minimize potential confounding factors and ensures

the reliability of the findings.

While the manuscript presents novel and important insights, there are certain areas that could

benefit from further clarification and improvement. Please find below my detailed comments and

suggestions.

Reply: we thank the reviewer for the positive comments.

Specific comments:

Issues to address:

  • In figure 1, could you show the gene expression results as fold change for peri-tumoral vs

tumoral regions of the tissue? Fold change is easy for the readers to interpret the results.

Reply: we could agree with the reviewer that the results expressed as fold change is easy for the readers to interpret. Unfortunately, considering the feedback from four other reviewers as well, we have decided to keep the presentation of the results unchanged, modifying only the graphical representation by using dot plots instead of histograms.

  • In figure 1, could you also add gene expression results for lcn-2 and mmp-9 of healthy vs

UADT cancer patients? This would be good to correlate the protein results in your figure 2.

Reply: The collection of tissues from healthy patients for gene expression analysis is prevented by ethical considerations, as the invasive procedure cannot be justified when not related to a medical necessity. Therefore, for tissue analysis, non-tumoral sections from cancer patients were used as “healthy” controls.

  • For table 3, instead of adding all the correlation p values into the table, could you add

correlation plots that would show the correlation between all the proteins that you are

testing? Visualization with the plots would be better for the readers instead of reading the

values in the table.

Reply: according to reviewer request, we added as supplementary results, the correlation plots of our data.

Round 2

Reviewer 1 Report

Comments and Suggestions for Authors

I am satisfied with the review of the manuscript.

Author Response

We sincerely appreciate the referee's suggestions, which have contributed to improving the manuscript.

Reviewer 2 Report

Comments and Suggestions for Authors

The authors have addressed my concerns.

But the similarity rate is still too high. The authors should reduce the rate before acceptance.

Author Response

We sincerely appreciate the referee's suggestions, which have contributed to improving the manuscript.

We have made an effort to reduce the similarity found in the Materials and Methods section. However, we believe that the redundancy regarding instrument codes and materials used for the analyses should be considered a valuable addition, ensuring transparency and replicability of the research.

Reviewer 3 Report

Comments and Suggestions for Authors

The authors modified the manuscript replying to some of the reviewer's queries.

Author Response

(The authors gave the same response as above.)

Reviewer 5 Report

Comments and Suggestions for Authors

Dear Authors

Revised manuscript explains well. 

Author Response

(The authors gave the same response as above.)
